

# Marine microbial communities of the Great Barrier Reef lagoon are influenced by riverine floodwaters and seasonal weather events

Florent E. Angly[1], Candice Heath[1], Thomas C. Morgan[1], Hemerson Tonin[2], Virginia Rich[3,4], Britta Schaffelke[2], David G. Bourne[2] and Gene W. Tyson[1]

[1] Australian Centre for Ecogenomics, University of Queensland, St Lucia, Queensland, Australia
[2] Australian Institute of Marine Science, Townsville, Queensland, Australia
[3] Department of Soil, Water and Environmental Science, University of Arizona, Tucson, AZ, United States of America
[4] Microbiology Department, Ohio State University, Columbus, OH, United States of America

Corresponding author
Florent E. Angly,
florent.angly@gmail.com

## ABSTRACT

The role of microorganisms in maintaining coral reef health is increasingly recognized. Riverine floodwater containing herbicides and excess nutrients from fertilizers compromises water quality in the inshore Great Barrier Reef (GBR), with unknown consequences for planktonic marine microbial communities and thus coral reefs. In this baseline study, inshore GBR microbial communities were monitored along a 124 km long transect between 2011 and 2013 using 16S rRNA gene amplicon sequencing. Members of the bacterial orders Rickettsiales (e.g., Pelagibacteraceae) and Synechococcales (e.g., *Prochlorococcus*), and of the archaeal class Marine Group II were prevalent in all samples, exhibiting a clear seasonal dynamics. Microbial communities near the Tully river mouth included a mixture of taxa from offshore marine sites and from the river system. The environmental parameters collected could be summarized into four groups, represented by salinity, rainfall, temperature and water quality, that drove the composition of microbial communities. During the wet season, lower salinity and a lower water quality index resulting from higher river discharge corresponded to increases in riverine taxa at sites near the river mouth. Particularly large, transient changes in microbial community structure were seen during the extreme wet season 2010–11, and may be partially attributed to the effects of wind and waves, which resuspend sediments and homogenize the water column in shallow near-shore regions. This work shows that anthropogenic floodwaters and other environmental parameters work in conjunction to drive the spatial distribution of microorganisms in the GBR lagoon, as well as their seasonal and daily dynamics.

# INTRODUCTION

Coral reefs are among the most biologically diverse and productive ecosystems on Earth. However, these complex assemblages, often compared to tropical rainforests, are
under increasing anthropogenic pressure. Reefs are experiencing a rapid decline due to a combination of local pressures such as overfishing, nutrient enrichment, increased land runoff and sedimentation, and global disturbances such as rises in temperature (*Pandolfi et al., 2003*; *De'ath et al., 2012*). The GBR is a World Heritage Area and the largest reef complex in the world, stretching over 2,100 km along the Queensland coast of Australia. Despite being considered one of the best managed marine areas, the GBR is exposed to nutrient, sediments and pollutant inputs from land-based activities (*Schaffelke et al., 2012a*; *Schaffelke et al., 2013*) resulting in a 50.7% decrease in coral cover over the last 27 years (*De'ath et al., 2012*). Given the fundamental socio-economic role coral reefs have in many countries (food production, tourism, coastal protection) and their ecological value (biodiversity and productivity), it is vital that these ecosystems are better understood and protected.

Microorganisms are a diverse group of unicellular organisms that form the base of the marine food chain (*Azam et al., 1983*), hence indirectly sustaining higher order organisms including invertebrates and fish. They are also an essential component of the coral holobiont, and disturbing the balance between the corals and their associated microbiota has been implicated in reduced reef health (*Dinsdale et al., 2008*; *Bruce et al., 2012*). In addition, the small size and fast reproduction rate of microorganisms make them very efficient at cycling nutrients, metabolizing foreign compounds in marine ecosystems and colonizing new ecological niches (*Thurber et al., 2009*).

In the GBR lagoon, river runoff from agricultural areas introduces sediments, excess nutrients from fertilizers (e.g., phosphate and nitrate) and pesticides (herbicides or insecticides) from the land to the inshore waters (*Furnas, 2003*; *Brodie et al., 2012*), predominantly during discrete, short-lived flood events during the 5-month summer monsoonal wet season. Land use changes over the past 200 years (increased agriculture, urbanization) have increased the amounts of sediments, nitrogen, phosphorus and herbicides in these floodwaters (*Devlin & Brodie, 2005*; *Devlin et al., 2012a*), with profound impacts on coastal ecosystems (*Schaffelke, Mellors & Duke, 2005*; *Fabricius, 2005*; *Brodie & Mitchell, 2005*; *De'ath & Fabricius, 2011*; *Schaffelke et al., 2013*). Particularly high levels of herbicides such as diuron are currently found in the GBR lagoon, which inhibits the photosystem II and damages mangroves, seagrass, corals, and other non-target photosynthetic organisms (*Lewis et al., 2009*; *Shaw et al., 2010*). While herbicides can be toxic to some microorganisms (*Leboulanger et al., 2008*), they can be neutral to others that have dedicated enzymes for their degradation (*Aislabie & Lloyd-Jones, 1995*). To date, microbial communities co-existing with the other macroscopic species on the GBR have not been characterized and it is unclear how anthropogenic compounds found in seasonal runoff affect these communities.

In this study, we characterized planktonic microbial communities of seven GBR lagoon sites differentially exposed to inputs from the rivers of the Wet Tropics catchment. Over three years, we determined water chemistry and characterized microbial communities using 16S rRNA gene amplicon sequencing. We hypothesized that microbial communities follow seasonal dynamics and respond to riverine input, potentially buffering reef
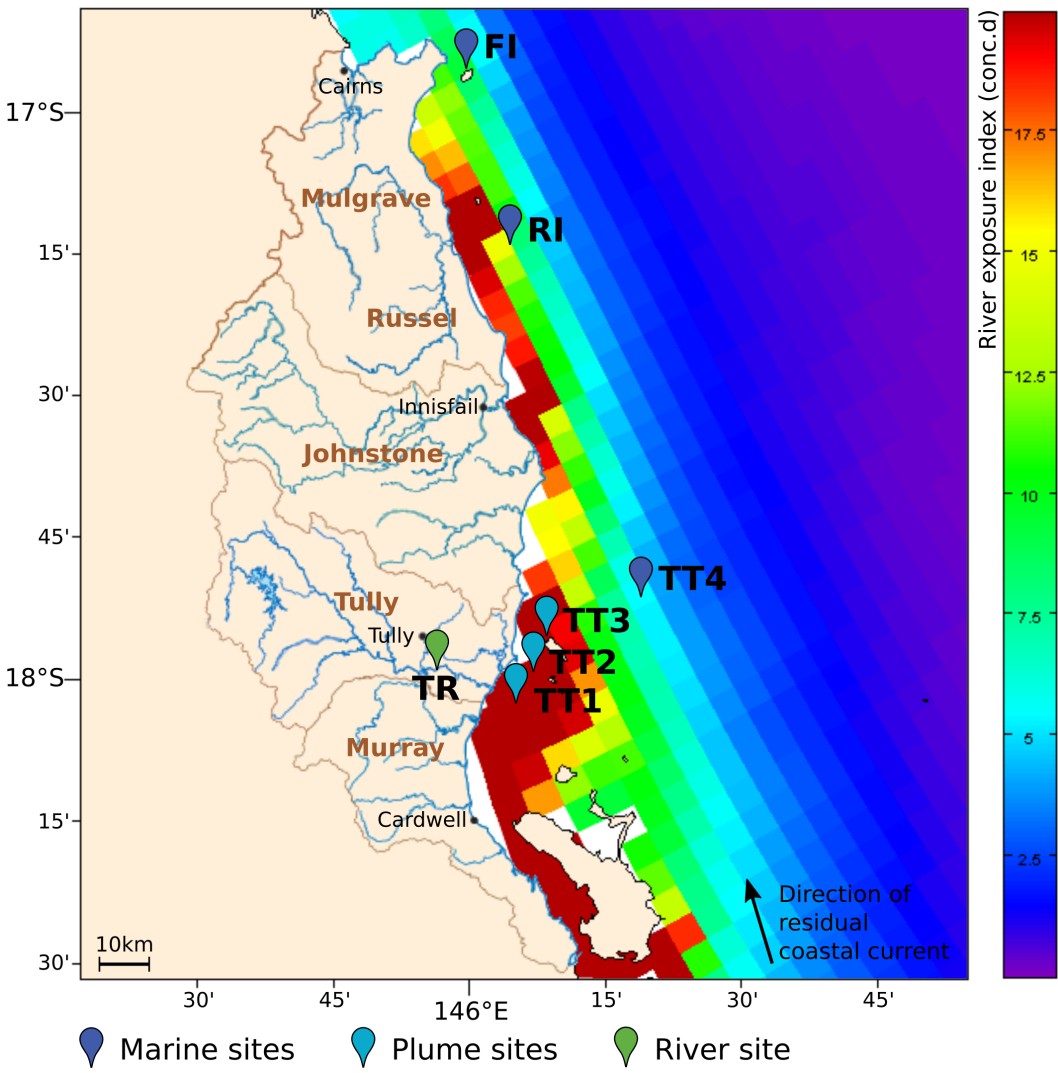

Figure 1 **Overview of sampling area in the GBR lagoon.** The river exposure index is shown for the Wet Tropics river catchments in the 2010–11 wet season, with a color bar indicating clustered cumulative exposure (concentration $x$ days) above 1% of the incoming concentration (capped at 20 conc.d). The direction of the residual coastal current is indicated as a black vector. The location of the sites surveyed for microbial composition in 2011–13 is shown as colored paddles. The sites were classified as marine, plume or riverine, according to their respective distance to the nearest influent river mouth.

ecosystems against effects of elevated floodwater constituents through nutrient cycling and detoxification.

## MATERIALS & METHODS

### Sampling design

Sampling was performed in the Wet Tropics Region of the GBR (Fig. 1), a well-studied coastal area which is regularly exposed to river runoff and flood events (*Devlin & Schaffelke, 2009*; *Schroeder et al., 2012*; *Turner et al., 2013*). The sites surveyed were located on a transect following a gradient of river exposure, from the highly-exposed Tully River

mouth (TT1) to the fringing coral reefs of Dunk Island (TT3) that are seasonally reached by flood plumes, and the TT4 off-shore location, rarely exposed to river water. Russel (RI) and Fitzroy islands (FI) were additional reef sites with limited exposure to the waters of the Johnstone and Russel rivers, respectively, and a consistently higher coral health index than Dunk Island (*Thompson et al., 2014*). All sampling sites were classified based on their proximity to the nearest influencing river mouth: 'plume' for <20 km downstream and 'marine' if >20km (Table S1). To characterize the influence of riverine microorganisms on marine communities, a freshwater site located 12 km upstream of the river mouth was also selected (TR).

The seven sites were surveyed between January 2011 and October 2013 in the dry season (June), just prior to the wet season (October), and at the end of the wet season (March). At each site and sampling date, a single 2 L seawater sample was taken from a depth of 5 m (just below surface for the TR site, which was a very shallow river bed), passed through a 0.22 μm Sterivex filter, which was stored at −20 °C until further processing. All samples were collected under the auspices of the general permit (G12/35236.1) granted by the Great Barrier Reef Marine Park Authority to the Australian Institute of Marine Science.

## Environmental conditions

Water samples collected from 2011 to 2013 were processed according to the long-term GBR Reef Rescue Marine Monitoring Program (*Thompson et al., 2013*) to assess temperature, salinity, bottom depth and water chemistry: concentrations of suspended solids (SS), particulate organic carbon (POC), particulate phosphorus (PP), particulate nitrogen (PN), dissolved inorganic nitrogen (DIN), silica (Si) and chlorophyll *a* (CHLA). In addition, diuron concentration was determined by collecting 1 L of water in pre-washed bottles, and storing the water at 4 °C until processing by solid phase extraction liquid chromatography tandem mass spectrometry (SPE-LC-MS/MS) at Queensland Health and Forensic Scientific Services, Coopers Plains, Australia. All these measured environmental parameters were deposited as NCBI BioSamples (accession # PRJNA276058). In addition, meteorological parameters were acquired from public resources: Tully River discharge and water temperature (Department of Natural Resources and Mines, http://watermonitoring.derm.qld.gov.au/host.htm), and solar exposure and rainfall (Bureau of Meteorology, http://www.bom.gov.au/climate/data/index.shtml). An average of these meteorological parameters was calculated for the seven days preceding each sampling date.

A river exposure index was calculated using a hydrodynamic model (http://www.bom.gov.au/environment/activities/coastal-info.shtml), based primarily on the Sparse Hydrodynamic Ocean Code (SHOC) hydrodynamic model (http://www.emg.cmar.csiro.au/www/en/emg/software/EMS/hydrodynamics.html). SHOC is a general purpose model, applicable on spatial scales ranging from estuaries to regional ocean domains (*Herzfeld, 2006*). We used outputs from the regional application of SHOC to the GBR using a horizontal spatial resolution of about 4 km, with a model grid size of 180 × 600 with 48 vertical layers with 1 m resolution at the surface. In this context, conservative tracers were

used in this study to simulate the transport of unique tracers 'released' from different rivers. In mathematical terms, a conservative substance is represented without terms of sinks or sources of mass in the transport equation. This means that the change in concentration values happens due to physical processes (advection and diffusion). Furthermore, it was stipulated that the tracers used do not affect the hydrodynamics. This technique enables the identification of marine regions influenced by individual catchments, and provides insight into the mixing and retention of river water along various regions in a given domain (*Brinkman et al., 2002*; *Brinkman et al., 2014*; *Luick et al., 2007*). Model simulations of the 3-dimensional distributions of passive tracers were analyzed to produce weekly estimates of cumulative exposure to tracers above a threshold of 1% of the source concentration. An exposure index was calculated that integrates the tracer concentrations above this threshold, based on a cumulative measurement of the exposure concentration and duration of exposure related to individual river sources, and expressed as Concentration x Days (conc.d). For every location in the model domain, the cumulative exposure index was calculated as:

$$\text{Conc.Days} = \sum_{t=0}^{T} \text{Conc}_{exceed} \times t \text{ where}$$

$$\text{Conc}_{exceed} = \begin{cases} \text{Conc}(t) - \text{Conc}_{thresh}, \text{ if } \text{Conc}(t) > \text{Conc}_{thresh} \\ 0, \text{ if } \text{Conc}(t) \leq \text{Conc}_{thresh} \end{cases}$$

and $\text{Conc}_{thresh}$ is defined as 1% of the source concentration, $\text{Conc}(t)$ represents the time-varying tracer concentration, and $t$ is the time in days from the beginning of the wet season to the end (01 November–31 March). Cumulative exposure was calculated for each grid point in the model domain. Using this representation, the exposure index integrates both concentration above a defined threshold and the duration of exposure. For example, an exposure of 20 days at a concentration of 1% above the threshold would produce an index value of 0.2, which is equivalent to 10 days exposure at 2% above the concentration threshold. This index provides a consistent approach to assess relative differences in exposure of inshore GBR waters to inputs from various rivers. For each of the wet seasons simulated by the model, spatial maps of river exposure indices were calculated for the target rivers: Herbert, Tully, Murray, Johnstone, Mulgrave and Russel rivers (Wet Tropics catchment), Burdekin and Haughton rivers (Burdekin catchment, affecting the south of the Wet Tropics catchment).

## 16S rRNA gene amplicon sequencing

DNA was extracted from each Sterivex filter using a modified method from *Suzuki et al. (2004)*. In brief, the filters were thawed on ice with Invitrogen's P1 buffer with lysozyme at a final concentration of 2 mg/mL, and incubated for 30 min at 37 °C, while rotating at 10 rpm. Proteinase K (0.75 mg/mL final concentration) and 10% sodium dodecyl sulfate (1% final concentration) were added and the sample was incubated, with rotation, at 55 °C for 2 h. DNA was extracted using phenol:choloroform:isoamyl alcohol (25:24:1; pH 8.0) followed by an overnight ethanol precipitation and purified using a MO BIO PowerClean DNA Clean-Up kit (Carlsbad, CA, USA).

Amplicons were generated by PCR-amplifying the V6–V8 variable regions of the 16S rRNA gene using the pyroLSSU926F and pyroLSSU926F universal primers as described in *Dove et al. (2013)*. The resulting DNA amplicons were sequenced on a Roche-454 GS-FLX instrument at the Australian Centre for Ecogenomics and deposited in the NCBI Short Read Archive (accession # PRJNA276058).

## Bioinformatic processing

Amplicon reads were processed using Hitman (https://github.com/fangly/hitman), a bioinformatic workflow based around the UPARSE methodology (*Edgar, 2013*). In brief, Hitman: (1) joins read pairs with PEAR (*Zhang et al., 2014*), but keeps the forward read when pairs cannot be joined; (2) truncates the 3′ end of sequences at the first residue below a threshold quality value ($Q$) using TRIMMOMATIC (*Bolger, Lohse & Usadel, 2014*); (3) trims the 3′ end of all sequences to a target length ($L$) using TRIMMOMATIC, discarding all smaller sequences, (4) removes sequences exceeding the maximum number of expected errors ($E$) using USEARCH's fastq_filter (*Edgar & Flyvbjerg, 2015*); (5) uses USEARCH's cluster_otus to form operational taxonomic units (OTUs) from high-fidelity sequences (stringent quality processing in steps 2 and 4) that are sorted by decreasing abundance, occur at least twice in the dataset and meet a minimum percentage of similarity ($O$); (6) discards chimeric OTUs using USEARCH's cluster_otus in a reference-independent, and using UCHIME (*Edgar et al., 2011*) based on a reference database ($C$); (7) assigns regular-fidelity sequences (less stringent quality processing in steps 2 and (4) to each OTU using USEARCH's usearch_global (*Edgar, 2010*); (8) formats the results in BIOM format using Bio-Community's bc_convert_files (*Angly, Fields & Tyson, 2014*); (9) gives a taxonomic assignment to each OTU by globally aligning their representative sequences against a database ($T$) of reference sequences trimmed to the target region (keeping only the best-matching alignment with a minimum required identity percentage ($I$) using USEARCH's usearch_ global; (10) removes OTUs belonging to specific taxa ($W$) using Bio-Community's bc_manage_ samples; (11) rarefies the microbial profiles at the given depth ($D$) with Bio-Community's bc_accumulate assuming an infinite number of bootstrap replicates; and (12) corrects gene-copy number bias using CopyRighter (*Angly et al., 2014*).

In this study, Hitman was run using the following parameters: $L = 250$ bp, $Q = 7$ (16 for HiFi sequences), $E = 3.0$ expected errors (0.5 for HiFi sequences), $O = 97\%$ identity (species-level), $C =$ GOLD database (*Bernal, Ear & Kyrpides, 2001*), $T =$ merged Silva (*Quast et al., 2012*) and Greengenes (*McDonald et al., 2012*) databases (https://github.com/fangly/merge_gg_silva), $I = 95\%$ identity (genus-level), $W =$ "Eukaryota**Chloroplast*"" and $D = 279$ for Bacteria & Archaea (100 for Eukaryotes). In addition, rarefaction curves were generated using Bio-Community (*Angly, Fields & Tyson, 2014*).

## Statistical analysis

All statistical analyses were carried out using the R language (*R Core Development Team, 2015*)). Comparisons of diversity between groups of samples were carried out using the non-parametric, unilateral Mann–Whitney $U$ test (wilcox.test() function).

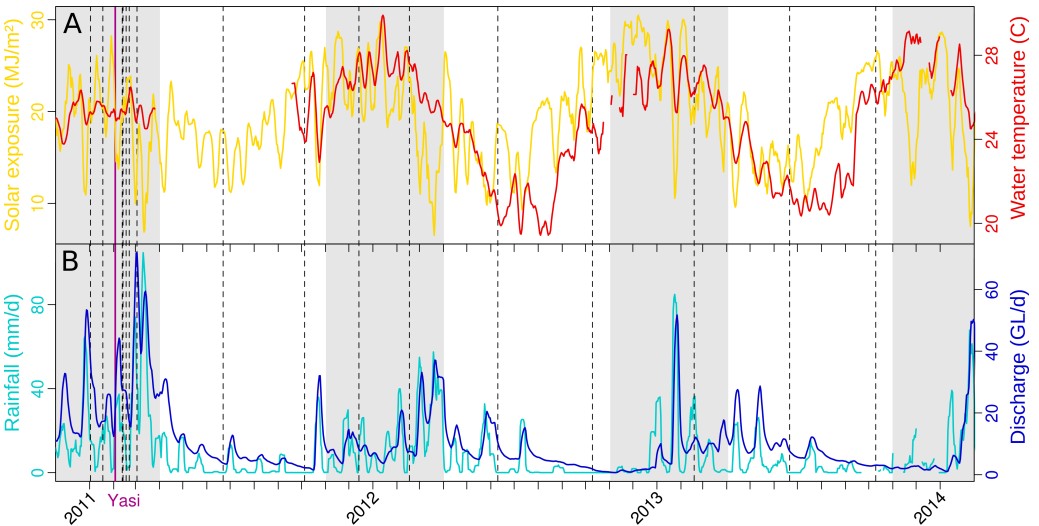

**Figure 2** **Weather in the Tully catchment during the years 2011–13.** (A) temperature and solar exposure, and (B) rainfall and river discharge. An average value for the previous week is reported for each day. Dashed lines indicate microbial sampling dates, and the purple line the landfall of tropical cyclone Yasi. The shading represents the extent of the wet season. Sources: BOM, DERM.

Principal coordinates analysis (PCoA) and PERMANOVA were performed using the capscale() and adonis() functions of the vegan package (*Dixon, 2003*). The indicspecies package (*Cáceres & Legendre, 2009*) was used to determine indicator species with the multipatt() function. Redundancy analysis (RDA) model selection was based on the AIC (Akaike information criterion) and calculated by ordistep() in vegan. Pearson correlations between environment variables were computed using rcorr() from the Hmisc package. The functions fa.parallel(), fa(), target.rot() and fa.diagram() from the psych package were used to conduct exploratory factor analysis (EFA), i.e., to identify groups of co-varying variables. EFA was performed on several subsets of the data including different environmental parameters and the results were summarized.

## RESULTS & DISCUSSION

### Sampling and environmental context

Seven inshore GBR sites exposed to different levels of river runoff from the Wet Tropics catchments were surveyed over three years for water chemistry assessments and determination of microbial community structure. The classification of these sites as plume or marine sites was based on their distance to the nearest influencing river mouth (Table S1) and matched well with their river exposure index as calculated by oceanographic modeling (Fig. 1); sites <20 km from a river mouth were more highly exposed to riverine water (>20 conc.d) than sites >20 km away (<15 conc.d).

The weather in the Tully catchment from 2010 to 2013 followed the expected seasonal dynamics, dry and cool conditions between the months of May and October, hot and humid with most of the annual rainfall from November to April (Fig. 2). However, the 2010–11 wet season was marked by extreme weather (Table S2) and the landfall

of category 5 tropical cyclone Yasi (3 February 2011), that significantly affected coral reefs (*Perry et al., 2014*). Annual river discharges reached a record high (*Schaffelke et al., 2011*), causing elevated exposure to nutrients, PSII herbicides and sediments across most inshore GBR regions (*Devlin et al., 2012b*; *Kennedy et al., 2012*; *Perry et al., 2014*). The following wet season (2011–12) represented a return to typical weather conditions, with river discharge close to the long-term median (*Schaffelke et al., 2012a*; *Wallace et al., 2014*). However, coral reef recovery was delayed until 2014, when the coral health index reached pre-2011 levels again (*Thompson et al., 2014*).

## Microbial diversity

The archaeal and bacterial microbial profiles obtained by Roche-454 sequencing of 16S rRNA gene amplicons were rarefied (279 counts per sample) to allow comparison of microbial profiles (Fig. S1). A ten times higher sequencing effort would have been needed to sample nearly all the OTUs present in these aquatic samples (richness; Fig. S1A). Nevertheless, the chosen rarefaction depth recovered the vast majority of archaeal and bacterial diversity (Shannon–Wiener index; Fig. S1B). The microbial diversity of the rarefied profiles was calculated, with a median richness of ∼65 OTUs in the river, and ∼90–100 at the marine and plume sites (Fig. 3A). At plume sites, richness (Fig. 3A) and evenness (Fig. 3B) were higher in the dry season than in the wet season, which corresponded to a larger overall diversity (Shannon–Wiener index) in the dry season (Fig. 3C) (Mann–Whitney $U$ test; $p < 0.05$).

## Establishing a microbial monitoring baseline for inshore GBR

Taxonomic assignments and gene copy number correction were conducted to produce accurate estimates of microbial relative abundance (*Angly et al., 2014*) (Fig. S2) that yield an understanding of the prevalence of microbial taxa in the GBR lagoon waters. At a coarse taxonomic level, the bacterial orders Sphingobacteriales, Burkholderiales, and Xanthomonadales dominated the TR river site (Fig. 4), while the archaeal order E2 and bacterial orders Rickettsiales and Synechococcales were prevalent at the plume and marine sites (Fig. 4). On average, Rickettsiales and E2 had higher relative abundance at the marine sites (29.6 and 17.7% respectively) compared to the plume sites (20.4 and 10.4 % respectively). This distribution was not constant over time, and notably, the plume sites were characterized by a large fraction of Burkholderiales in the January–March 2011 period, i.e. the extreme wet season 2010–11. Flavobacteriales were found in all sites, riverine, plume and marine. Burkholderiales are Betaproteobacteria commonly found in rivers (*Cottrell et al., 2005*; *Ghai et al., 2011*; *Liu et al., 2012*). Sphingobacteriales have been reported at riverine locations affected by waste water treatment effluent (*Drury, Rosi-Marshall & Kelly, 2013*), where they may degrade complex compounds such as herbicides and antibiotics (*Kämpfer, 2011*). Synechococcales such as those found in the plume and marine sites include the *Synechococcus* and *Prochlorococcus* genera, which represent the main photosynthetic bacteria in oceanic waters (*Partensky, Blanchot & Vaulot, 1999*; *Partensky, Hess & Vaulot, 1999*), and the small heterotrophic Rickettsiales are also commonly reported in the ocean (*Morris et al., 2002*; *Carlson et al., 2008*). More specifically,

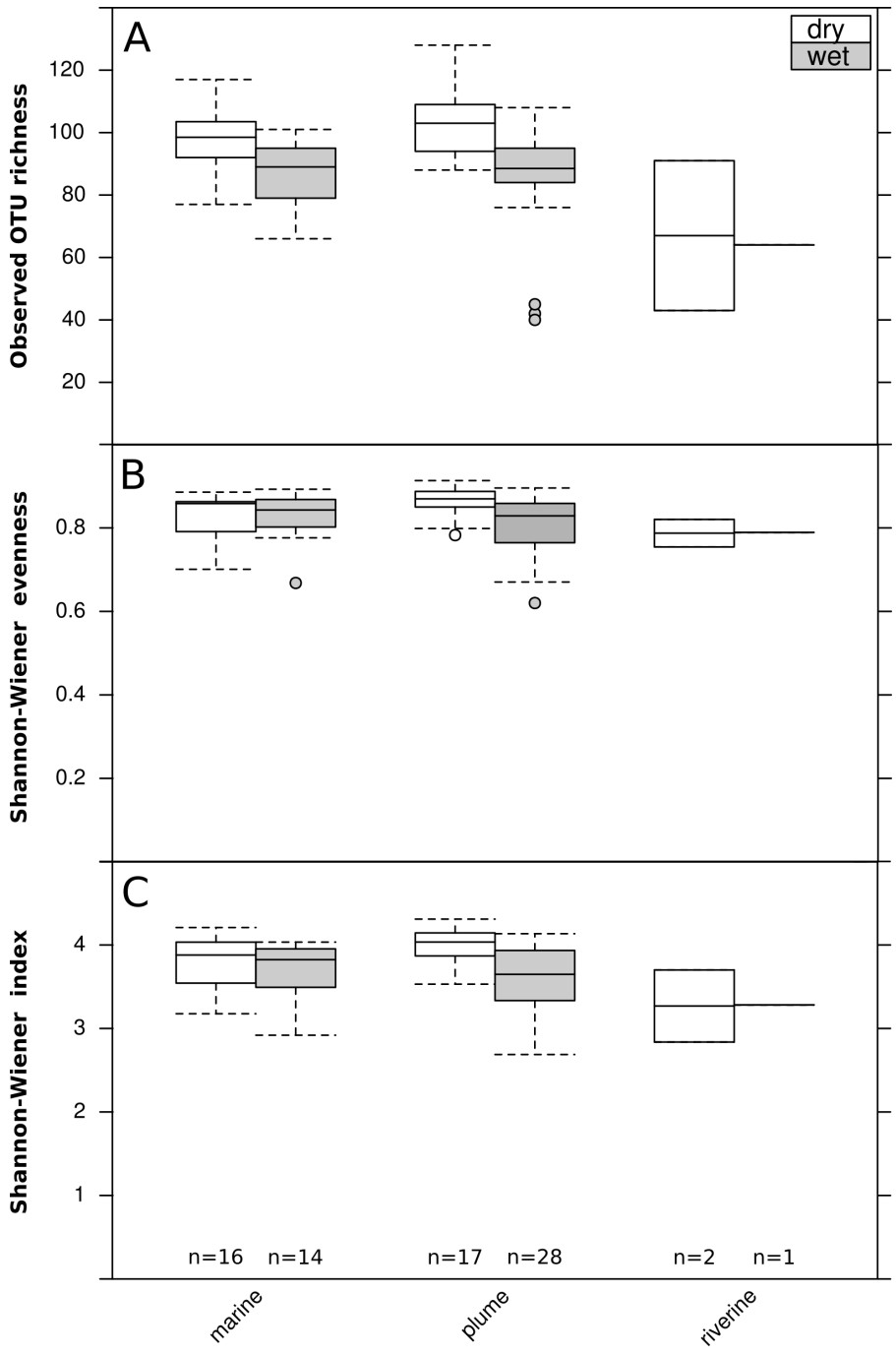

**Figure 3** **Seasonal and spatial differences in microbial diversity.** Boxplot of seasonal (wet or dry) and spatial (three runoff exposure categories) differences in bacterial and archaeal diversity: (A) observed OTU richness, (B) Shannon-Wiener evenness, (C) Shannon-Wiener index. Boxes represent the first quartile, median and third quartile of the data, whiskers the minimum and maximum, and circles the outliers. Blue bars show the statistical comparisons performed and significant differences are represented by a star (Mann–Whitney $U$ test; $p < 0.05$).

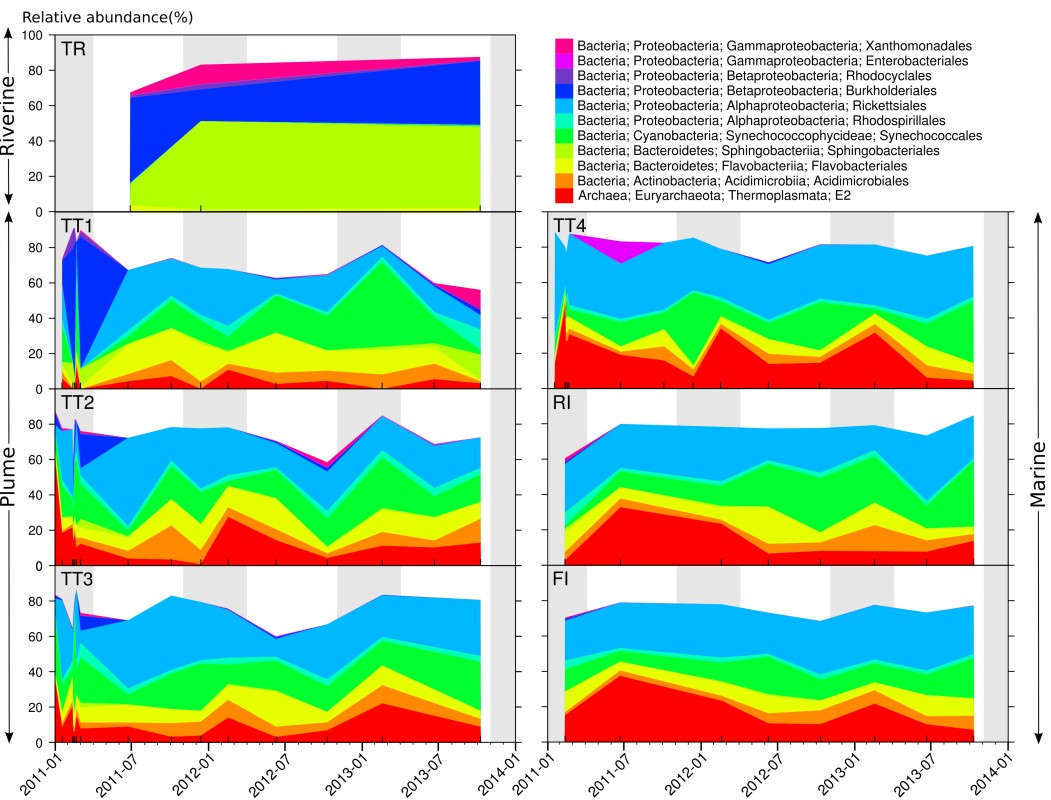

**Figure 4** **Distribution and dynamics of abundant microbial orders for the seven sites surveyed in 2011, 2012–2013.** Microbial orders present at less than 10% relative abundance in all samples were omitted. The shading represents the extent of the wet season.

both orders were previously identified in coral reef waters (*Kelly et al., 2012*; *Lu et al., 2015*). However, Thermoplasmata E2 Archaea have only occasionally been reported in temperate surface seawater (*Massana et al., 1997*; *Pernthaler et al., 2002*). These Archaea are motile photo-heterotrophic cells focused on the degradation of protein and lipids (*Iverson et al., 2012*) and are found in coral mucus, suggesting a potential role in maintaining coral health (*Kellogg, 2004*).

Genomic methods such as 16S rRNA gene amplicon sequencing are effective for marine monitoring (*Bourlat et al., 2013*). The present taxonomic characterization of the microbial communities in inshore GBR waters represents a baseline against which future microbiological studies can be compared. This baseline may prove valuable for assessing future change in this reef ecosystem, be it further degradation or recovery.

## Geographical and temporal distribution of OTUs

Non-constrained ordination (PCoA) was applied to get a precise account of the dynamics and distribution of specific microbial taxa in inshore GBR. At the finer operational taxonomic unit (OTU) level (Fig. 5A), significant spatial (distance to river mouth and site type; PERMANOVA, $p < 0.05$) and temporal effects (wet or dry; PERMANOVA, $p < 0.05$) were confirmed. Site type (riverine, plume or marine) had the strongest effect (27.8%

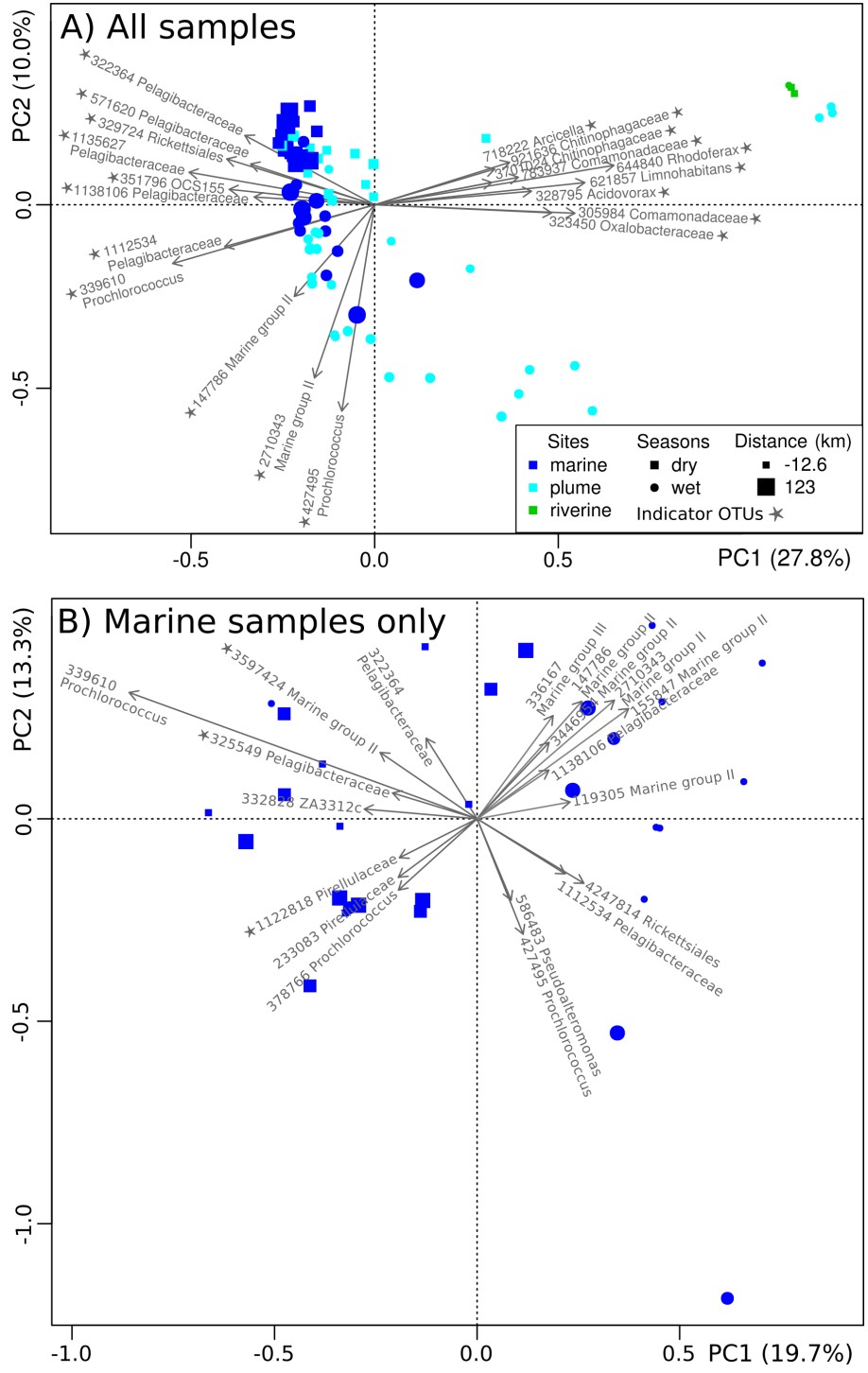

**Figure 5** **Spatiotemporal organization of microbial communities.** PCoA plot showing the spatiotemporal organization of microbial communities based on their Hellinger dissimilarity. Symbol color, shape and size depend on the type of site, season of collection and distance from the Tully River mouth, respectively. The OTUs explaining the largest variation are represented by arrows and their assigned Greengenes ID and genus-level taxonomy is shown. A star indicates indicator OTUs (indicator species analysis; $p < 0.05$).

explained variance along PC1, Fig. 5A), followed by seasons (wet or dry) (10.0% explained variance along PC2, Fig. 5A).

Marine and plume samples were characterized by indicator OTUs, i.e. OTUs characteristic of the marine and plume areas (indicator species analysis; $p < 0.05$). These indicator OTUs belonged to the Rickettsiales (e.g., Pelagibacteraceae), Synechococcales (e.g., *Prochlorococcus*), Acidimicrobiales (e.g., OCS155), and the Archaeal order E2 (e.g., Marine Group II) (Fig. 5A), which is largely consistent with the trends observed at the order level (Fig. 4). The main difference between the OTU- and order-level analyses (Figs. 5 and 4 respectively) was the detection of OCS155, an OTU indicator of marine and plume sites (Fig. 5A). OCS155 belongs to the Acidimicrobiales, a group of likely planktonic, free-living microorganisms with a photo-heterotrophic lifestyle (*Mizuno, Rodriguez-Valera & Ghai, 2015*) found in all tropical and temperate photic areas (*Ghai et al., 2013*).

Indicator OTUs for the river samples were also identified (indicator species analysis; $p < 0.05$), including members of the orders Burkholderiales (family Comamonadaceae, e.g. *Limnohabitans*, *Acidovorax* and *Rhodoferax*; family Oxalobacteraceae), and Sphingobacteriales (family Flexibacteraceae, e.g., *Arcicella*; family Chitinophagaceae) (Fig. 5A). These results are congruent with those at the order level (Fig. 4). The presence of riverine OTUs belonging to the order Burkholderiales and Sphingobacteriales at some plume sites surveyed in the wet months (Figs. 4 and 5A), when river discharge is high, is a sign that these sites are affected by riverine water effluent.

When repeating the PCoA with the marine samples only, i.e. restricting the analysis to samples unaffected by river output, a clear partition between dry and wet seasons (Fig. 5B) was seen (PERMANOVA, $p < 0.05$). All samples clustered in one of two season-specific groups arranged along PC1, except for one sample collected at TT4 in December 2011. Several indicator OTUs were identified (indicator species analysis; $p < 0.05$) (Table S3), e.g., Pelagibacteraceae (3 OTUs), Pirellulaceae (2 OTUs) and Marine group II (2 OTUs), which were specific of the dry season. Generally though, Marine group II OTUs seemed to be more abundant in the wet season (Figs. 5A and 5B). Previous studies have shown seasonality in near-shore microbial communities (*Treusch et al., 2009*; *Gilbert et al., 2009*; *Gilbert et al., 2012*). In particular, *Pelagibacter* is known to exhibit seasonality (*Alonso-Sáez et al., 2007*; *Carlson et al., 2008*; *Eiler et al., 2009*; *Fortunato et al., 2013*). Overall, the microbial communities of the GBR lagoon seem to respond to seasonal influence, although it is not as pronounced as the influence of geographical location, an observation that was also made in a previous investigation of river to ocean gradient (*Fortunato et al., 2012*).

In addition to a seasonal effects, GBR lagoon samples were susceptible to short term effects of potentially high magnitude. Rapid community changes were evident when looking at the Tully transect samples (TT1–4) collected in the wake of Cyclone Yasi (03 Feb 2011) (Fig. S3). The influence of Tully river input was marked, with elevated levels of Sphingobacteriales and Burkholderiales recorded between 13 and 17 February. While these changes were especially pronounced at the Tully River mouth, even the offshore site TT4 experienced analogous changes in this period. Especially large changes in community structure were observed between consecutive days, on February 12–13 (Fig. 2). These findings add to previous investigations of soil and gut microbiota, which
have shown that community composition can change within a few days (*Schmidt et al., 2007*; *Michelland et al., 2011*; *David et al., 2014*). While baselines may be established using e.g., monthly sampling, it is clear that future research should adopt an intensive daily sampling regimen to better characterize such transient changes in estuarine and marine microbial communities.

## Environmental drivers of coastal microbial dynamics

A range of measurements describing the environmental context of the microbial samples were collected in this study, including temperature, salinity, bottom depth and water chemistry variables (concentrations of suspended solids, silica, chlorophyll *a*, the herbicide diuron, dissolved and particulate form of carbon, nitrogen and phosphorus), and complemented with weather condition data (Tully River discharge, local rainfall and solar exposure). Many of these environmental parameters were highly correlated (Pearson test; 48.6% of the pairs with $r > 0.3$), requiring the need for EFA to extract groups of uncorrelated parameters. Following this procedure (Fig. S4), four independent factors were identified. Factor MR1 included local rainfall and Tully River discharge. Factor MR2 included chlorophyll *a*, suspended solids and other particulates (POC, PN, PP). Factor MR3 included DIN and salinity and can be interpreted as mixing with freshwater. Factor MR4 combined water temperatures and solar exposure. Although larger discharge (factor MR1) can lead to increased suspended solids (factor MR2) and salinity decrease (factor MR3), factor MR1, MR2 and MR3 were not directly correlated because salinity and the amount of suspended solids depend not only on site location (proximity to the river mouth) but also on the action of wind and waves, which homogenize the water column and resuspend sediments (*Larcombe et al., 1995*; *Orpin & Ridd, 2012*; *Fabricius et al., 2014*). A water quality index based on comparison to water quality guidelines (*GBRMPA, 2010*) and ranging from −1 for poor quality to +1 for very good quality was previously introduced (*Thompson et al., 2013*). This index aggregates scores given to four indicator concentrations (suspended solids, chlorophyll *a*, particulate nitrogen, particulate phosphorus), which are all part of the suspended material factor (factor MR2) identified by EFA in the present study. Diuron has a strong association with sediments (*Stork, Bennett & Bell, 2008*; *Balakrishnan, Takeda & Sakugawa, 2012*; *Xu et al., 2013*) and is also included in this factor. Overall, this suggests that factor MR2 can be interpreted as the quality of the water.

Constrained ordinations (RDA) were carried out to study the relative importance of the four independent factors (MR1–4) on microbial community structure. A single environmental parameter was chosen to represent each factor prior to conducting RDA with model selection: rainfall for factor MR1, water quality index for MR2, salinity for MR3, and water temperature for MR4 (Fig. 6). These four environmental parameters were significantly associated with microbial community composition (PERMANOVA, $p < 0.05$). Microbial community structure was affected by temperature (Fig. 6A), consistent with previous findings in the Western English Channel (*Gilbert et al., 2009*), which could explain the seasonality identified in marine sites. Further, seasonality can be attributed to the higher rainfall typical of the wet season (Fig. 6A). However, large changes could be attributed to decreases in water quality index and salinity, especially for plume sites

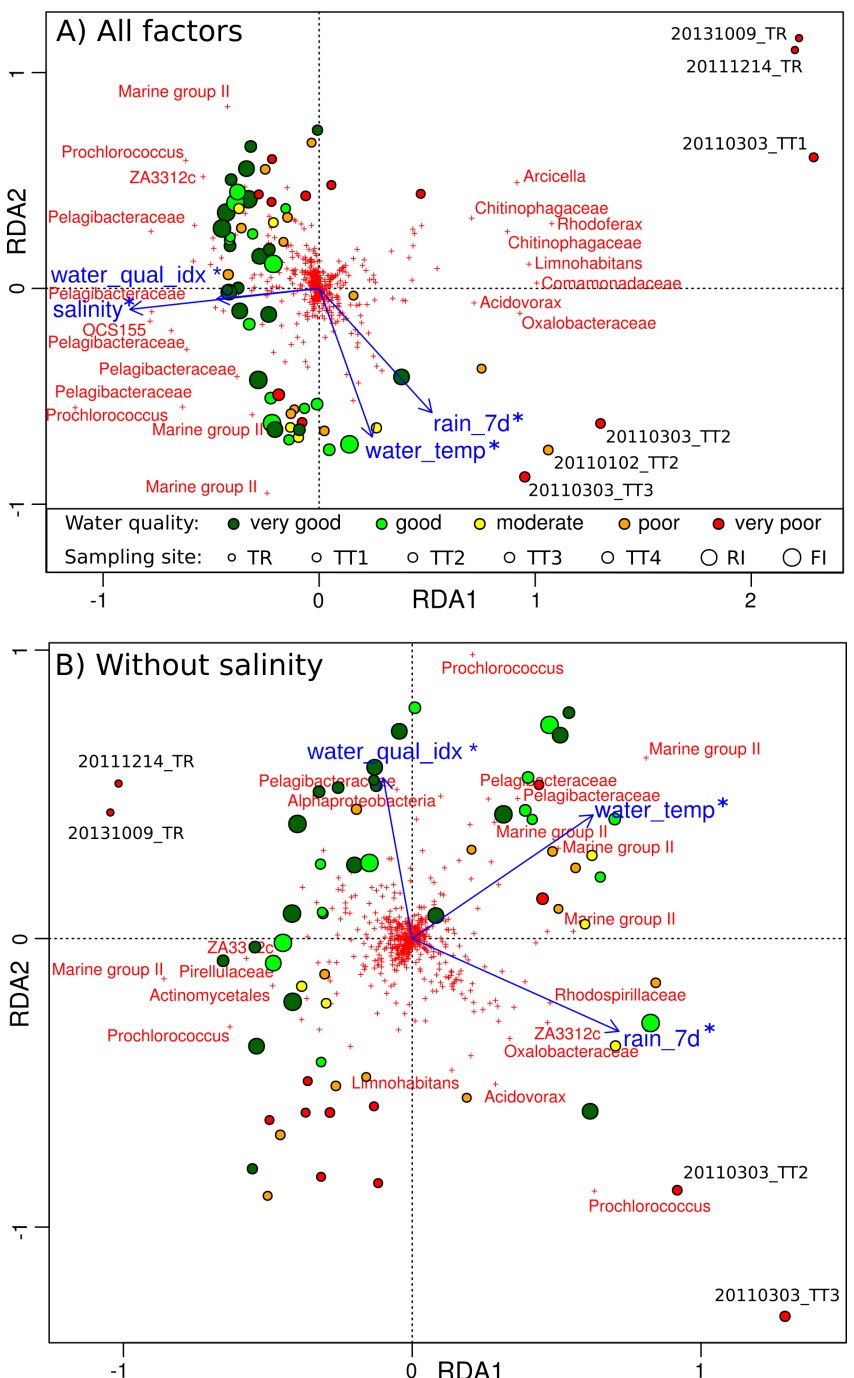

**Figure 6 Relationship between environmental parameters and microbial profiles.** RDA plot showing the relationship between GBR environmental parameters and Hellinger-transformed microbial profiles: (A) for all environmental factors, and (B) with the contribution of salinity removed. Disks represent sampling sites and are colored according to the AIMS water quality index (dark green: very good, green: good, yellow: moderate, orange: poor, red: very poor). OTUs are depicted by red crosses and the genus-level Greengenes taxonomy of the most discriminating ones is shown. The factors explaining sample distribution are represented by blue arrows: rainfall in the last 7 days (rain_7d), water temperature (water_temp), water quality index (water_qual_idx) and salinity. Any significant association is indicated by a star (PERMANOVA; $p < 0.05$).

(Fig. 6A), with negative consequences for corals (*Fabricius et al., 2014*). These changes can be interpreted as the effect of riverine floodwater, which was especially pronounced for sites located near the Tully river mouth, and helps explain the geographical pattern seen in microbial communities. The effects of salinity and water quality index were mostly collinear in Fig. 6A, despite the fact that they represent two separate factors (MR2 and MR3). Repeating the RDA analysis with the influence of salinity removed (conditioning term, Fig. 6B), water quality index was still identified as a significant driver of microbial communities (PERMANOVA, $p < 0.05$). This suggests that, as a whole, nutrient, organic compounds and herbicides such as diuron brought by riverine water into the ocean have an effect on microbial communities.

RDA analysis was also conducted to identify the factors driving the large, transient changes detected by microbial profiling at the plume sites on 12–13 February 2011 (Fig. S5). Being only nine days after the landfall of tropical cyclone Yasi, this period was characterized by extreme weather, e.g., high river discharge (12 Feb: 37.1 ML/d; 13 Feb: 36.2 ML/d; 7–13 Feb mean: 27.5 ML/d). While river discharge was similar on the two days, the February 13 microbial communities were characterized by elevated levels of suspended solids and lower salinity (PERMANOVA, $p < 0.05$), suggesting that other environmental factors may contribute to microbial community structure. We speculate that larger wind or waves were present on February 13, which would have decreased the vertical stratification of the water column, resulting in lower salinity and higher concentration of suspended solids. This example illustrates that salinity and suspended solids at inshore locations may be a function of both river flow and wind (*Schaffelke et al., 2012b*; *Fabricius et al., 2014*). The 12 February was also marked by an increase in the eukaryotic to prokaryotic reads ratio (EPR) (Fig. S6), with a maximum EPR of 1.45 reached on February 13 at TT1. The most abundant eukaryal taxon in this sample was assigned to the hydrozoan genus Merona, that includes very small organisms that can spawn eggs (*Schuchert, 2004*). These data could be interpreted as the potential spawning of hydrozoan or the displacement from their usual habitat (e.g., the benthic zone) and they illustrate that extreme weather may dramatically change environmental conditions, thereby affecting all microbial kingdoms (Archaea, Bacteria and Eukaryota).

## CONCLUSIONS

This study is a baseline description of microbial communities in the inshore GBR lagoon. Marine Group II Archaea, Pelagibacteraceae and Rickettsiales were prevalent in all the seawater samples. A seasonal effect of temperature and rainfall on the microbial communities was apparent in the three year sampling period. However spatial effects were more pronounced, with sites located close to the Tully river mouth including many river-specific taxa, particularly during the wet season. Seasonal storms like those that occurred in the wet season 2010–11 caused elevated suspended solids and decreased salinity at plume sites, which translated into large, transient changes in microbial community structure. Water quality played a role in driving microbial community structure in the GBR lagoon, but the complex interconnections between environmental parameters mean that future

research such as experimental manipulations will be needed to precisely elucidate how each individual anthropogenic compound shapes microbial community composition and affects coral reefs.

## ACKNOWLEDGEMENTS

The authors would like to thank the AIMS water quality monitoring team and the crew of the research vessel RV Cape Ferguson for their support during field work, as well as Fiona May from the Australian Centre for Ecogenomics for her assistance with 454 gene amplicon sequencing.

### Funding

This study was supported by the Australian Research Council's Discovery Early Career Research Award to Florent Angly (DE120101213) and a Queen Elizabeth II fellowship to Gene Tyson (DP1093175). The fieldwork was carried out during sampling cruises for the Marine Monitoring Program, which was supported by the Great Barrier Reef Marine Park Authority, through funding from the Australian Government's Caring for our Country Program and the Australian Institute of Marine Science. The funders had no role in study design, data collection and analysis, decision to publish, or preparation of the manuscript.

### Grant Disclosures

The following grant information was disclosed by the authors:
Australian Research Council's Discovery Early Career Research Award: DE120101213.
Queen Elizabeth II fellowship: DP1093175.
Australian Government's Caring for our Country Program and the Australian Institute of Marine Science.

### Competing Interests

The authors declare there are no competing interests.

### Author Contributions

- Florent E. Angly conceived and designed the experiments, performed the experiments, analyzed the data, wrote the paper, prepared figures and/or tables, reviewed drafts of the paper.
- Candice Heath performed the experiments, analyzed the data, reviewed drafts of the paper.
- Thomas C. Morgan performed the experiments, reviewed drafts of the paper.
- Hemerson Tonin performed the experiments, prepared figures and/or tables, reviewed drafts of the paper.
- Virginia Rich conceived and designed the experiments, reviewed drafts of the paper.
- Britta Schaffelke and David G. Bourne conceived and designed the experiments, performed the experiments, contributed reagents/materials/analysis tools, reviewed drafts of the paper.

- Gene W. Tyson conceived and designed the experiments, performed the experiments, reviewed drafts of the paper.

## Field Study Permissions

The following information was supplied relating to field study approvals (i.e., approving body and any reference numbers):

All samples were collected under the auspices of the general permit (G12/35236.1) granted by the Great Barrier Reef Marine Park Authority to the Australian Institute of Marine Science.

## DNA Deposition

The following information was supplied regarding the deposition of DNA sequences:

NCBI Short Read Archive PRJNA276058.

## Data Availability

All raw data collected in this study (DNA sequences and environmental measurements) were deposited in the NCBI databases under the accession number PRJNA276058.

## Supplemental Information

Supplemental information for this article can be found online at http://dx.doi.org/10.7717/peerj.1511#supplemental-information.

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
