# Peer review of "Marine microbial communities of the Great Barrier Reef lagoon are influenced by riverine floodwaters and seasonal weather events"

_PeerJ, doi:10.7717/peerj.1511_

## Round 0.1 · original submission · Minor Revisions

· Academic Editor

Minor Revisions

Dear Authors,

I agree with the reviewers' comments. And additional comments are below:

1. The authors call R and the library as a package. Usually, R is a language and the library is the package.
2. Explain how the statistical analysis help to answer the biological questions of the manuscript. This should be better explained in the text.
3. Clarify the definition of water quality variable in the M&M.
4. There is a disbalance between the number of marine and riverine samples. How can it interfere in the analysis?

Please, make sure you address all the Editor and reviewers' suggestions or requirements, point-by-point.

Reviewer 1 ·

Basic reporting

Angly et al. performed a broad diversity study in river, plume, and sea at Tuly river area. The authors use both 16S rRNA pyrosequencing and physical-chemical characterization of water quality. They went on and found correlations between environmental features and microbial types, proposing indicator microbes and environmental indexes. I have only a few minor remarks.

1. How did the authors manage to differentiate Synechos from Prochloros using the 16S fragment (Line 251, 272)?
2. I did not find a discussion on the seasonality of the plume (Line 311). Is it always covering the same marine area? If not, how it changes over the time?
3. Remove/re-work Lines 260-264.
4. Can the authors discuss further the health index/water quality in a broader context comparing with other reef areas (line 314)?

Experimental design

.

Validity of the findings

.

Additional comments

.

Reviewer 2 ·

Basic reporting

The research paper “Marine microbial communities of the Great Barrier Reef lagoon are influenced by riverine floodwaters and seasonal weather events” by Angly et al., investigates bacterial and archaeal diversity in the vicinity of the Great Barrier Reef using 16S rDNA amplicon pyrosequencing spanning a three year timeline. The presented results are a good baseline description of the water quality parameter and its association with bacterial and archaeal richness. Although the manuscript is technically sound, it needs improvement regarding the discussion of the data obtained, which is quite descriptive in nature. I have several comments to be addressed by the authors.

Experimental design

The experimental design is acceptable, although I have some comments on the General comments items 1 – 2.

Validity of the findings

The data and the data analysis are strong and sound which provided a reasonable first step towards a baseline for the water quality and bacterial richness. However, I have concerns about the impact of archaeal results given the experimental set up (item 3 on General Comments).

Additional comments

Some suggestions are provided below.
1) I could not find in the Material and Methods section whether each site sampled has biological replicates.
2) It is stated that a 10 times greater sequencing effort would be needed to cover/saturate the OTUs richness. Why did the authors use pyro instead of illumina sequencing to achieve just that?
3) I am sure there are now available primer sets more suitable to target the Archaea domain (that are based on the alignment of a large number of archaea) than the ones listed on M&M. Why the authors use the same set to target both Bacteria and Archaea?
4) A better explanation must be added for the selection points used. For instance, all points seem to follow the shoreline (even TT4, RI, and FI) where the disturbance from the main land river and plumes, probably, would be the more striking. The question would be why not add a transector moving away from the mainland/ river plumes to address how far this influence takes place both in terms of water quality and microbial richness fluctuations.
5) Figure 6 in gray scale is very difficult to read, please change font to black.
6) It would be interesting to the readers to have some information about the coral health on the closest reefs during the sampling time and location points. The water quality components were well monitored during the process, but nothing is reported about the status of coral health. Would the plumes, at least in the wet season, be strong enough to influence coral health status?
7) Line 366 and Figure S6 in the Results/Discussion section “This unusual extreme event period…eukaryotic to prokaryotic reads ratio (EPR)…” the authors want to use EPR to make the case for water column disturbances? If so, this statement does very little for the discussion, unless the authors want to establish some kind of ecological meaning this paragraph is not needed.

---

## Round 0.2 · accepted · Accept

· Academic Editor

Accept

Based on the revised version of the manuscript, I have decided that your manuscript can be accepted for publication.